# Special Low Protein Foods Prescribed in England for PKU Patients: An Analysis of Prescribing Patterns and Cost

**DOI:** 10.3390/nu13113977

**Published:** 2021-11-08

**Authors:** Georgina Wood, Alex Pinto, Sharon Evans, Anne Daly, Sandra Adams, Susie Costelloe, Joanna Gribben, Charlotte Ellerton, Anita Emm, Sarah Firman, Suzanne Ford, Moira French, Lisa Gaff, Emily Giuliano, Melanie Hill, Inderdip Hunjan, Camille Newby, Allison Mackenzie, Rachel Pereira, Celine Prescott, Louise Robertson, Heidi Seabert, Rachel Skeath, Simon Tapley, Allyson Terry, Alison Tooke, Karen van Wyk, Fiona J. White, Lucy White, Alison Woodall, Júlio César Rocha, Anita MacDonald

**Affiliations:** 1Faculty of Health, Education & Life Sciences, Birmingham City University, City South Campus, Westbourne Road, Edgbaston, Birmingham B15 3TN, UK; 2Dietetic Department, Birmingham Women’s and Children’s Hospital, Steelhouse Lane, Birmingham B4 6NH, UK; alex.pinto@nhs.net (A.P.); evanss21@me.com (S.E.); a.daly3@nhs.net (A.D.); anita.macdonald@nhs.net (A.M.); 3Royal Victoria Infirmary, Queen Victoria Road, Newcastle upon Tyne NE1 4LP, UK; sandraadams2@nhs.net; 4Royal Devon & Exeter NHS Foundation Trust, Barrack Rd, Exeter EX2 5DW, UK; s.costelloe@nhs.net; 5Guy’s and St Thomas’ NHS Foundation Trust, London SE1 7EU, UK; Joanna.Eardley@gstt.nhs.uk (J.G.); Sarah.Firman@gstt.nhs.uk (S.F.); 6University College London Hospitals NHS Foundation Trust, National Hospital for Neurology & Neurosurgery, Queen Square, London WC1N 3BG, UK; c.ellerton@nhs.net; 7University Hospital Southampton NHS Foundation Trust, Tremona Road, Southampton SO16 6YD, UK; anita.emm@uhs.nhs.uk; 8North Bristol NHS Trust, Southmead Road, Bristol BS10 5NB, UK; suzanne.ford@nbt.nhs.uk; 9University Hospitals of Leicester NHS Trust, Infirmary Square, Leicester LE1 5WW, UK; moira.french@uhl-tr.nhs.uk; 10Cambridge University Hospitals NHS Foundation Trust, Hills Road, Cambridge CB2 0QQ, UK; lisa.gaff@addenbrookes.nhs.uk (L.G.); celine.prescott@addenbrookes.nhs.uk (C.P.); 11Northamptonshire Healthcare NHS Foundation Trust, St Mary’s Hospital, London Road, Kettering NN15 7PW, UK; emily.giuliano@nhft.nhs.uk; 12Sheffield Teaching Hospitals NHS Foundation Trust, Herries Road, Sheffield S5 7AU, UK; melanie.hill13@nhs.net; 13Bradford Teaching Hospitals NHS Foundation Trust, Duckworth Lane, Bradford BD9 6RJ, UK; Inderdip.Hunjan@bthft.nhs.uk; 14Bristol Royal Hospital for Children, Bristol BS2 8BJ, UK; Camille.Newby@uhbw.nhs.uk; 15Royal Derby Hospital, Uttoxter Road, Derby DE22 3NE, UK; allison.mackenzie1@nhs.net; 16Norfolk and Norwich University Hospital, Colney Lane, Norwich NR4 7UY, UK; rachel.pereira@nnuh.nhs.uk; 17University Hospitals Birmingham NHS Foundation Trust, Queen Elizabeth Hospital, Birmingham B15 2TH, UK; Louise.robertson@uhb.nhs.uk; 18Somerset NHS Foundation Trust, Parkfield Drive, Taunton TA1 5DA, UK; heidi.seabert@somersetft.nhs.uk; 19Great Ormond Street Hospital for Children NHS Foundation Trust, Great Ormond Street, London WC1N 3JH, UK; rachel.skeath@gosh.nhs.uk; 20University Hospitals Bristol & Weston NHS Foundation Trust, Marlborough St, Bristol BS1 3NU, UK; simon.tapley@uhbw.nhs.uk; 21Alder Hey Children’s NHS Foundation Trust, E Prescot Road, Liverpool L12 2AP, UK; allyson.terry@alderhey.nhs.uk; 22Nottingham University Hospitals NHS Trust, Queen’s Medical Centre Campus, Derby Road, Nottingham NG7 2UH, UK; alison.tooke@nuh.nhs.uk; 23Royal Manchester Children’s Hospital, Oxford Road, Manchester M13 9WL, UK; karen.vanwyk@cmft.nhs.uk (K.v.W.); fiona.white@cmft.nhs.uk (F.J.W.); 24Sheffield Children’s NHS Foundation Trust, Clarkson St, Broomhall, Sheffield S10 2TH, UK; lucy.white19@nhs.net; 25Salford Royal NHS Foundation Trust, Stott Lane, Salford M6 8HD, UK; alison.woodall@srft.nhs.uk; 26Nutrition & Metabolism, NOVA Medical School, Faculdade de Ciências Médicas, Universidade Nova de Lisboa, Campo Mártires da Pátria 130, 1169-056 Lisbon, Portugal; rochajc@nms.unl.pt; 27CINTESIS—Center for Health Technology and Services Research, NOVA Medical School, Campo Mártires da Pátria 130, 1169-056 Lisbon, Portugal; 28Reference Centre of Inherited Metabolic Diseases, Centro Hospitalar Universitário de Lisboa Central, 1169-045 Lisbon, Portugal

**Keywords:** special low protein foods, phenylketonuria, England, prescribing patterns, costs

## Abstract

Patients with phenylketonuria (PKU) are reliant on special low protein foods (SLPFs) as part of their dietary treatment. In England, several issues regarding the accessibility of SLPFs through the national prescribing system have been highlighted. Therefore, prescribing patterns and expenditure on all SLPFs available on prescription in England (*n* = 142) were examined. Their costs in comparison to regular protein-containing (*n* = 182) and *‘free-from’* products (*n* = 135) were also analysed. Similar foods were grouped into subgroups (*n* = 40). The number of units and costs of SLPFs prescribed in total and per subgroup from January to December 2020 were calculated using National Health Service (NHS) Business Service Authority (NHSBSA) ePACT2 (electronic Prescribing Analysis and Cost Tool) for England. Monthly patient SLPF units prescribed were calculated using patient numbers with PKU and non-PKU inherited metabolic disorders (IMD) consuming SLPFs. This was compared to the National Society for PKU (NSPKU) prescribing guidance. Ninety-eight percent of SLPF subgroups (*n =* 39/40) were more expensive than regular and *‘free-from’* food subgroups. However, costs to prescribe SLPFs are significantly less than theoretical calculations. From January to December 2020, 208,932 units of SLPFs were prescribed (excluding milk replacers), costing the NHS £2,151,973 (including milk replacers). This equates to £962 per patient annually, and prescribed amounts are well below the upper limits suggested by the NSPKU, indicating under prescribing of SLPFs. It is recommended that a simpler and improved system should be implemented. Ideally, specialist metabolic dietitians should have responsibility for prescribing SLPFs. This would ensure that patients with PKU have the necessary access to their essential dietary treatment, which, in turn, should help promote dietary adherence and improve metabolic control.

## 1. Introduction

Phenylketonuria (PKU), an inborn error of amino acid metabolism, is caused by phenylalanine hydroxylase deficiency, an enzyme that converts phenylalanine to tyrosine [1]. This leads to neurotoxicity, causing severe intellectual disability if untreated [2]. It is managed by a life-long phenylalanine-restricted diet supplemented with a phenylalanine free/low phenylalanine protein substitute, although adjunct pharmacological therapies may also be prescribed to some patients [2,3]. In particular, patients with classical PKU require severe restrictions of natural protein, commonly tolerating ≤25% of a normal protein intake [1,2]. Regular protein containing foods e.g., bread, flour and pasta, are replaced with special low protein foods (SLPFs) that contain minimal protein [2,3]. These deliver a substantial source of energy, providing up to 50% of daily energy intake [4,5,6], fibre [7], they offer essential bulk, add variety and so help to sustain dietary adherence and ultimately aid metabolic control [8,9,10].

The cost of SLPFs to patients in England is reimbursed by the National Health Service (NHS), as these foods are considered borderline substances and are available on NHS prescription [11,12,13]. Borderline substances are nutritional or dermatological products specifically formulated to manage a medical condition [12]. There are around 150 SLPFs available on borderline substance prescription in England [13]. Each SLPF is approved by the United Kingdom (UK) Advisory Committee on Borderline Substances (ACBS) [12,13,14], which considers the clinical need of a product, its efficacy and the total price to the NHS [15]. Manufacturers/suppliers of SLPFs provide the ACBS with a statement outlining the proposed NHS list price and any distribution costs charged to dispensers [15]. For SLPFs that are broadly similar to existing products, the ACBS recommends a maximum benchmark cost to the NHS for that category [15]. When a company chooses to increase their NHS list price and maintain ‘ACBS status’, price increases are benchmarked against a standard inflation comparator [15].

General Practitioners (GPs) issue prescriptions for SLPFs monthly on request, which are then dispensed through local pharmacists or specialist home delivery companies linked to the suppliers of SLPFs [16]. The NHS then pays pharmacists or dispensing doctors a fee for each item they dispense [17,18]. The National Society for PKU (NSPKU) has produced a guide outlining the maximum monthly number of units of SLPFs (e.g., 1 unit = 1 pack of pasta up to 500 g—see Appendix A for full list of definitions for each product) which can be prescribed [19,20]. This guide considers patient age and circumstances to support GPs in prescribing these products and to ensure that expenditure on SLPFs is controlled. This guide has been widely adopted by GPs. In England, NHS prescriptions are free of charge for patients in the following categories: under 16 years of age; aged 16–18 years if in full time education; over 60 years of age; pregnant; receive income support or in other specific circumstances [21]. All other patients must pay a set fee per item, or they can purchase a three-monthly or annual prescription prepayment certificate which covers all of their NHS prescriptions [21].

However, there are many challenges in accessing SLPFs with the current prescribing system [16,22]. Some patients with PKU report that they have had their prescription requests refused; some describe how their GPs advise that they should purchase these foods rather than obtain them on prescription [16]. Others report that their GPs refuse to prescribe the appropriate range of products, as they consider some foods luxury items (e.g., cake mix or cereal bars) or the quantity of SLPFs is reduced due to their costs [16]. In a study by MacDonald et al., 2019, 43% (*n =* 25/58) of caregivers and parents said they needed more SLPFs for their children than they had been prescribed [22]. These challenges will impact on nutritional intake, directly affecting nutritional status and ultimately metabolic control.

Although studies have considered the cost of SLPFs, the majority were conducted outside the UK, where different reimbursement systems exist [23,24,25,26]. One study compared the theoretical costs in 10 international centres, where costs of SLPFs in the UK appeared to be higher than in many other countries [11]. Two nonpeer reviewed articles also discussed the theoretical cost of SLPFs in the UK and suggested that some SLPFs are expensive, but emphasised they are essential in the management of PKU [27,28]. Several papers have discussed costs when looking at the challenges of living with PKU in the UK, but this has not been the single focus of their work [3,16,22,29,30]. No study has compared the costs of SLPFs with regular foods or foods used in other therapeutic diets. Furthermore, no study has considered the prescribing pattern of SLPFs for low protein diets in England, or the UK as a whole.

This study therefore aimed to:(1)examine the cost of all SLPFs on NHS prescription in England and compare these with similar regular equivalent protein containing and ‘*free-from*’ dietary foods available in the supermarkets; and(2)determine NHS expenditure on SLPFs and examine the number of SLPF units prescribed annually in England

## 2. Materials and Methods

### 2.1. Cost of SLPFs in England in Comparison to Regular Foods and ‘Free-From’ Foods

Data was collected from August to October 2020 on the price of all individual SLPFs available on ACBS prescription in England using British National Formulary (BNF) resources (Website, mobile phone app and book) and from the following suppliers or manufacturers websites if prices were stated:Promin—https://prominpku.com/shop (accessed on 3 October 2020) [31]Taranis—https://prominpku.com/shop (accessed on 3 October 2020) [31]Metax—https://prominpku.com/shop (accessed on 3 October 2020) [31]

When individual prices of items were unavailable or unclear, companies were contacted directly via email. The cost per kg of each SLPF was calculated. SLPFs were divided into 40 subgroups of equivalent food product types, e.g., low protein burgers, sausages, cookies/biscuits, cake mixes. The mean and range costs across subgroups of similar products were calculated.

The mean and range cost per kg were collected and calculated for at least two regular protein-containing comparable foods and at least two ‘*free-from*’ comparable foods, from major supermarkets in England with data available online (ASDA, Morrisons, Sainsburys, Tesco, Waitrose, Ocado and Marks & Spencer). A ‘*free-from*’ food was defined as a food made without one or more specific ingredients, designed for people with food allergies or other intolerances/diseases e.g., coeliac disease. If data was unavailable from a supermarket’s website, it was obtained from alternative online shops or directly from the manufacturer. Where prices differed between supermarkets for the same regular protein-containing food or ‘*free-from*’ food, the mean value was recorded. Percentage differences between SLPFs and regular/*’free-from’* food subgroups for all mean costs were determined. Variations within ± 10% were considered comparable.

### 2.2. NHS Prescribing Patterns for SLPFs and Expenditure in England

One of the authors (A.P.) was given approval to access and extract prescribing data about SLPFs from the NHS Business Service Authority (NHSBSA) ePACT2 (electronic Prescribing Analysis and Cost Tool 2) for the costs and quantity of SLPFs prescribed in total and for each subgroup in England. This tool provided access to prescription data from the NHSBSA from January to December 2020. An ePACT2 bespoke training session was arranged with NHSBSA to ensure that all data was obtained and interpreted correctly. NSPKU prescribing guidance describing the definition of one unit for each SLPF was used to calculate the number of units of SLPFs prescribed in total and for each subgroup (Appendix A) [19,20].

In order to estimate the number of patients with PKU cared for by NHS centres in England, all NHS centres known to treat and monitor PKU patients were contacted in order to determine the number of patients with PKU (paediatric and adult), the number on dietary treatment (defined as those receiving prescribed protein substitutes and therefore potentially SLPFs), the number of shared care patients and the number of non-PKU inherited metabolic disorders (IMD) patients accessing SLPFs. Information was supplied by dietitians working in *n =* 26 NHS England hospitals/centres who care for patients with PKU. These data were used to calculate how many units of SLPFs were being prescribed per patient per month and the cost to the NHS per patient per month in England. This was then compared to NSPKU prescribing guidance.

## 3. Results

### 3.1. SLPFs, Regular Foods and Free-From Foods Costing Comparison

One hundred and forty-six SLPFs were identified as being available on ACBS prescription in England, with these products grouped and further subcategorised for comparison with at least two regular food products per subgroup. Regular and ‘*free-from*’ comparators for four SLPFs (Calogen neutral, Calogen banana, Calogen strawberry and Duocal—Nutricia) were unavailable. Thus, 142 SLPFs were available for comparison with 182 regular products and 135 ‘*free-from*’ products. Table 1 displays all SLPF, regular product and ‘*free-from*’ food subgroups (*n* = 40), the mean cost per kg of products within each subgroup and % differences between costs.

Sixty-eight of 142 SLPFs (48%) were unavailable on BNF resources at the time of data collection (August to October 2020), and therefore, their costs had to be obtained directly from the manufacturer or supplier’s website or through email contact with the manufacturer/supplier.

When analysed by subgroup, all SLPFs were more expensive than regular foods and ‘*free-from*’ foods, except for regular eggs and *‘free-from’* flavour puddings, where their cost per kg was comparable to low protein equivalents.

Low protein crispbread crackers, Xpots (low protein equivalent of a pot noodle) and milk replacements (liquid) had the highest percentage cost difference, being 1117% to 1143% more expensive than the regular food comparator. When compared to *‘free-from’* foods, low protein flour, bread mix and egg whites had the highest percentage differences (575% to 825%) in costs. In contrast, low protein milk powder, fish substitute and jelly were only 27% to 61% more expensive than their *‘free-from’* food comparators. Basic SLPFs, including bread, pasta, rice, noodles and milk replacers (liquid), were 76% to 451% more expensive than ‘*free-from*’ equivalent foods.

### 3.2. NHS Prescribing and Costing Data in England for SLPFs

Table 2 displays the prescribing and costing data for SLPFs from January–December 2020.

In total, 208,932 units of SLPFs (monthly mean of 17,451 units) were prescribed from January to December 2020. This equated to a total actual cost of £2,151,973 (monthly mean cost of £179,566). The most frequently prescribed subgroups were bread, pasta/rice and flour, in total equating to 54.6% of all SLPFs prescribed. Milk replacers accounted for the highest percentage (30.5%) of the total actual cost of these products. There is not a definition for a unit of milk replacer, as the amount prescribed should be determined on an individual patient basis (Appendix A) [19,20]. Flour, pasta/rice and bread each accounted for just over 10% of total actual cost of SLPFs from January to December 2020 (11.1%, 13.7% and 10.8%, respectively).

Other expenses included payment for containers, consumables and out of pocket expenses, contributing 4.4% (£94,669) of the annual SLPFs costs to the NHS in England. Out of pocket expenses reimbursed to the pharmacy may include: postage and packaging costs; handling costs; and the cost of phone calls to manufacturers or suppliers to order products [32]. Payment at a rate of 10p for every prescription item is paid for containers where the quantity of a prescription item is ordered outside of the pack size or a multiple of the pack size (except for those granted ‘special container status’ where it is not practical to split a pack) [33]. An additional payment of 1.24p is made for all prescriptions including SLPFs in case additional consumables may need to be dispensed by the pharmacist (e.g., oral syringes, measuring spoons), although SLPFs usually do not need additional consumables. [33]. Also, a dispensing fee of £1.29 is allocated for each item prescribed [18].

### 3.3. NHS Patient Prescribing and Costing Data for SLPFs in England Compared to NSPKU Guidelines

Patients with PKU are the major consumers of SLPFs. It is estimated that there were 2359 patients with PKU in hospital follow-up in England (1436 adult patients, 923 paediatric patients), with *n* = 1814 (77%) on dietary treatment (Table 3). There were a further 422 patients using SLPFs with other inherited metabolic disorders of protein metabolism in England, suggesting that approximately 2236 patients in total were accessing SLPFs. On average, 93 units were prescribed per patient per year, which equates to approximately 8 units per month per patient. This is significantly less than the recommended maximum number of units per patient that could be prescribed each month as outlined by the NSPKU (Table 4). Actual cost data suggest that it costs a monthly mean of £80 per patient.

For the 877 paediatric patients with PKU on full or partial diet, it was estimated that 20% were aged 4 months–3 years (*n* = 175), 20% 4–6 years (*n* = 175), 20% 7–10 years (*n* = 175) and 40% 11–18 years (*n* = 352). Therefore, if all of these children, combined with adults with PKU on a full or partial diet (*n* = 937) were receiving the maximum number of low protein items on prescription each month, as per NSPKU guidance (Table 4), this would equate to 77,575 units each month. This is much higher than the average monthly prescribed units of 17,451 (excluding milk replacers) for the calendar year of 2020.

## 4. Discussion

This is the first study to examine the cost of all SLPFs available on prescription in England compared to regular and *‘free-from’* foods available in supermarkets. It is also the first study to examine the number and type of low protein items prescribed and expenditure on individual SLPFs and total SLPFs prescribed by the NHS in England over 1 year. There is a lower than expected volume of SLPFs prescribed in England, meaning that the costs to prescribe these products are significantly less than theoretically calculated [11,28], with a total of 17,451 units per month, costing £179,566. This equates to an estimated annual cost to the NHS per person with PKU in England of £962 with just 8 units (excluding low protein milk) prescribed per person per month, indicating that patients are receiving significantly less than the upper NSPKU prescribing guidance [16,19,20].

Over half (54.6%) of the units of SLPFs prescribed from January to December 2020 were basic foods such as bread, flour/mixes and pasta/rice. This accounted for just over one-third (35.6%) of the total annual costs. Just under a third (30.5%) of the costs were attributed to prescribing special low protein milks (liquid). It is likely that it is primarily children accessing SLPFs, as recent research suggested that it is mainly children aged <10 years with PKU who use prescribed special low protein milks [6]. There was previous concern that there may be over prescription of sweet SLPFs [8]. In Scotland, a 2014 survey found that special low protein pasta/rice/couscous, biscuits and flour were most commonly ordered by children, whereas adults with PKU mainly ordered pasta/rice/couscous, flour and bread [8]. In contrast, the amount of special low protein snacks and desserts (*n* = 14/40 subgroups including low protein chocolate, cookies, biscuits, cakes, and crisps) prescribed in England was minimal, with each subgroup only accounting for 0.1–5.9% of all SLPFs prescribed and contributing just 0.1–3.0% of the total NHS expenditure on SLPFs from January to December 2020. This is consistent with research reporting that special low protein cakes, biscuits and chocolate provide minimal contributions to daily energy intake in children with PKU [6]. It is clear that the expenditure on prescribing SLPFs is limited, particularly for sweet foods.

Overall, very little is known about SLPFs usage by adults with PKU in England. Our study suggests that 35% of adults with PKU were not following a phenylalanine restricted diet (Table 3). Although some adult patients may use SLPFs, others may not attempt to access them due to the complexity of the access system or the costs of the prescription fee for every food item ordered, unless the individual is entitled to free prescriptions. In one UK survey, 15% of patients with PKU stated that recurrent access problems with SLPFs was frustrating, and even led them to abandon their dietary treatment [16]. GP administration staff have been described as unhelpful, judgemental or obstructive when ordering SLPFs [8,16]; home delivery services are complex and sometimes unreliable, and SLPFs may arrive out of date or damaged, or of poor quality [16]. Some children with PKU were not on dietary treatment or not accessing SLPFs; this was associated with mild PKU, a higher natural protein tolerance, using sapropterin as an adjunct therapy, young infants not yet on solids or a dislike of SLPFs.

It is understandable that SLPFs cost more than regular and *‘free-from’* foods. The demand for SLPFs is small in a limited global market. Few companies manufacture or distribute SLPFs in the UK [13]. Production runs are small scale with high staffing ratios, leading to increased costs. Some of the raw ingredients and packaging materials are purchased in low volumes, increasing productions costs. Packaging may be subject to frequent label changes due to alterations in legislation. Manufacturing wastage may be high if final products do not meet the necessary standards. Manufacturers also need to make some profit to allow them to invest in research and development to improve and expand their SLPF range.

The availability, accessibility and cost of SLPFs vary between countries [5,7,8,11,13,23,24,25,34]. Comparisons are challenging due to differences in currency, age of patients, degree of dietary adherence and study methodology. China reported a mean cost of $573 (American dollars or approximately £415) a year per patient for SLPFs [25], whereas the United States of America found a mean cost of $1615 (approximately £1171) for children aged 0–17 years for SLPFs and just $967 (approximately £701) for adults [23]. The Netherlands reported a mean annual cost of €680 (approximately £576) on SLPFs, whereas the Czech Republic found this value to be significantly higher at €1560 (approximately £1321) [24,26].

The overall use of SLPFs is affected by the national access system and any consequential economic burden [11,23,24,25,26]. Some countries do not reimburse SLPFs costs; but may be funded by insurance coverage [11,24]. When national reimbursement schemes do not exist, families have to self-finance the purchase of SLPFs [11,23,25,26]. This is a huge financial burden for patients, which influences their ability to adhere to dietary treatment [11,23,25,26].

For patients with PKU to have better access to SLPFs through the NHS, several recommendations should be implemented. Consistent with previous suggestions by MacDonald et al. and Ford et al. [16,22], specialist metabolic dietitians should play a key role in prescribing SLPFs, as they control dietary management and oversee any dietary changes according to the individual patient’s metabolic control, nutritional needs, growth and overall nutritional status. This would be more efficient, minimise administration time and professional and patient confusion and enable patients with PKU to have minimal contact with healthcare professionals/prescribers who know very little about their condition and how it is managed. Instead, their SLPF prescriptions would be managed by those who are most equipped to support them in meeting their dietary needs and maintaining good metabolic control.

This study has some limitations. When obtaining the cost of each SLPF in August–October 2020, 68 products were not visible on any BNF resource, and therefore, prices were obtained directly from the manufacturer or supplier of SLPFs. The selection of protein-containing foods and *‘free-from’* foods as comparators, and how the products were grouped, was subjective. Certain powdered/dried SLPF products e.g., burger mix, had to be compared to a prepared regular protein-containing or ‘*free-from*’ product e.g., cooked burger; therefore, the cost of the SLPF in its prepared form per kg was estimated. This study only examined products accessible on prescription in England compared with protein-containing products and *‘free-from’* foods available from supermarket websites in England. Also, NHS prescribing and costing data were only available for England and not the whole of the UK, and were only collected from January to December 2020. From March 2020 onwards, England experienced multiple ‘lockdowns’ due to the coronavirus pandemic, and it is possible that this may have affected food behaviours and, consequently, the number and/or types of SLPFs that patients were requesting on prescription. However, there was no evidence from clinical practice that use or supplies of SLPFs were affected in England.

When calculating the number of units of SLPF and the costs per person with PKU in England, the numbers of patients on dietary treatment were estimated. However, dietetic colleagues throughout England provided representative and recent data from their clinics. It is difficult to state exactly how many patients were requesting SLPFs, as we did not examine individual prescribing data for each patient. On ePACT2, there were nine occasions in 2020 where a SLPF appeared on a prescription, but the quantity prescribed was unclear. Consequently, these data were removed from our spreadsheet. It is possible that there may be under-reporting of SLPFs by the NHSBSA ePACT2. The NHSBSA ePACT2 trainers/help team stated that there was a small possibility that data can be incorrectly processed, but that data is scanned from each prescription form directly, so the NHSBSA ePACT2 should accurately reflect all the prescriptions issued in England.

## 5. Conclusions

The annual cost to the NHS in England to prescribe SLPFs is £962 per patient with PKU and non-PKU IMD conditions. Surveys have repeatedly shown that patients or caregivers have access difficulties with current systems. If patients with PKU are expected to adhere to their dietary treatment for life, they must be able to easily access all SLPFs on prescription in a timely manner via the NHS. Given how little is currently being spent on prescribing SLPFs in England in comparison to the upper NSPKU guidance, cost should not be given as a reason to restrict a patient’s access to their essential dietary treatment. A review of how SLPFs are prescribed, supplied and controlled is warranted to improve the system, which, in turn, could lead to increased dietary adherence and improved patient outcomes.

## Figures and Tables

**Table 1 nutrients-13-03977-t001:** Cost of low protein, regular and *‘free-from’* food products for each subgroup and the % differences between costs.

Subgroup	SLPFs		Regular Protein-Containing Foods		‘*Free-From*’ Foods		% Difference between SLPFs and Regular Foods	% Difference between SLPFs and *‘Free-From’* Foods
	*n*	Cost (£/kg)	*n*	Cost (£/kg)	*n*	Cost (£/kg)		
** *Bread/pizza bases* **								
**Bread**	*12*	11.11	*12*	2.67	*11*	6.30	316%	76%
		(8.23–16.13)		(1.31–5.00)		(3.27–11.40)		
**Pizza base**	*1*	19.80	*2*	5.17	*2*	9.93	283%	99%
				(4.00–6.33)		(9.86–10.00)		
** *Pasta/rice/noodles* **								
**Pasta/rice/**	*33*	15.28	*23*	2.60	*16*	3.65	488%	319%
**noodles**		(8.80–19.10)		(1.20–5.04)		(1.20–7.50)		
**Pasta and sauces (prepared)**	*5*	16.16	*10*	2.61	*6*	9.36	519%	73%
		(8.82–26.25)		(1.11–4.98)		(7.50–13.32)		
**Risotto**	*1*	22.00	*2*	6.82	*2*	7.50	223%	193%
				(6.49–7.14)		(7.50–7.50)		
**Xpots/pot noodles**	*4*	92.50	*8*	7.44	*4*	24.32	1143%	280%
		(92.50–92.50)		(4.00–9.09)		(16.67–40.32)		
** *Flour/mixes* **								
**Bread mix**	*1*	11.96	*2*	1.64	*2*	1.72	629%	595%
				(1.28–2.00)		(1.69–1.75)		
**Cake mix**	*4*	15.64	*4*	4.27	*4*	6.95	266%	125%
		(13.94–19.36)		(1.20–5.29)		(4.57–9.97)		
**Flour/All Purpose Mix**	*5*	14.80	*2*	1.37	*2*	1.60	980%	825%
		(11.90–18.02)		(1.21–1.54)		(1.50–1.70)		
**Pancake/**	*1*	15.33	*2*	5.14	*2*	8.34	198%	84%
**waffle mix**				(5.00–5.28)		(7.00–9.68)		
** *Egg/replacers* **								
**Egg (prepared)**	*3*	3.01	*2*	3.24	*2*	1.46	-7%	106%
		(1.89–4.08)		(2.46–4.02)		(1.36–1.55)		
**Egg whites (powder)**	*1*	108.10	*2*	49.92	*2*	16.02	117%	575%
				(40.00–59.83)		(15.00–17.04)		
** *Milk/replacers* **								
**Milk (liquid)**	*5*	5.84	*2*	0.48	*2*	1.06	1117%	451%
		(4.05–6.75)		(0.48–0.48)		(0.59–1.53)		
**Milk (powder)**	*1*	22.38	*2*	7.64	*2*	17.56	193%	27%
				(5.89–9.39)		(15.16–19.96)		
** *Meat/replacers* **								
**Burgers (prepared)**	*3*	16.88	*4*	6.04	*4*	7.44	179%	127%
		(8.82–20.91)		(5.02–7.35)		(4.02–10.00)		
**Fish (prepared)**	*1*	18.07	*2*	10.03	*2*	11.78	80%	53%
				(8.25–11.81)		(11.67–11.88)		
**Sausages**	*3*	23.72	*6*	5.10	*4*	8.47	365%	180%
**(prepared)**		(23.72–23.72)		(3.06–6.88)		(6.67–9.26)		
** *Breakfast and cereal bars* **								
**Breakfast bar**	*4*	42.08	*8*	11.30	*4*	13.83	272%	204%
		(42.08–42.08)		(6.42–15.28)		(8.57–18.18)		
**Breakfast cereal (dry)**	*3*	23.35	*6*	4.84	*6*	6.18	382%	278%
		(23.07–23.92)		(2.37–6.17)		(4.50–10.56)		
**Fruit bar**	*1*	37.60	*2*	6.63	*2*	13.23	467%	184%
				(4.28–8.99)		(11.25–15.20)		
**Hot breakfast cereal (dry)**	*4*	25.11	*4*	10.61	*4*	11.45	137%	119%
		(25.00–25.45)		(6.00–20.52)		(8.33–14.55)		
** *Snacks* **								
**Biscuits/**	*7*	43.37	*10*	8.03	*8*	10.39	440%	317%
**cookies**		(33.60–68.52)		(1.05–25.00)		(6.50–17.86)		
**Breadsticks**	*1*	41.87	*2*	8.20	*2*	14.69	411%	185%
				(5.60–10.79)		(12.76–16.62)		
**Cake**	*3*	26.00	*2*	5.58	*2*	13.04	366%	99%
		(26.00–26.00)		(5.41–5.75)		(11.58–14.49)		
**Chocolate**	*2*	52.32	*2*	7.62	*2*	12.08	587%	332%
		(49.10–55.54)		(7.44–7.81)		(11.30–12.86)		
**Crackers**	*3*	25.38	*6*	7.07	*4*	12.58	259%	102%
		(24.00–26.07)		(3.25–9.56)		(12.00–13.81)		
**Crisps**	*4*	37.50	*8*	8.46	*4*	16.05	343%	134%
		(37.50–37.50)		(6.67–10.33)		(14.71–17.39)		
**Crispbread crackers**	*1*	32.80	*2*	2.66	*2*	8.93	1133%	267%
				(1.33–3.98)		(8.89–8.98)		
**French toast crackers**	*1*	20.00	*2*	6.35	*2*	11.24	215%	78%
				(6.25–6.45)		(10.80–11.67)		
**Hazelnut spread**	*1*	35.43	*2*	5.16	*2*	10.55	587%	236%
				(2.88–7.43)		(9.30–11.80)		
** *Desserts* **								
**Dessert pot**	*2*	20.30	*4*	5.99	*2*	7.71	239%	163%
		(20.30–20.30)		(4.69–7.14)		(2.93–12.50)		
**Flavoured pudding (powder)**	*4*	30.68	*7*	9.79	*2*	29.00 (13.00–45.00)	213%	6%
		(30.68–30.68)		(6.65–11.43)				
**Jelly (unprepared)**	*2*	25.59	*2*	4.18	*2*	15.88	512%	61%
		(25.59–25.59)		(4.16–4.19)		(15.88–15.88)		
**Rice pudding**	*4*	24.35	*6*	3.26	*2*	8.25	647%	195%
		(24.35–24.35)		(2.17–3.86)		(8.00–8.50)		
**Yogurt**	*1*	7.19	*2*	2.68	*2*	3.13	168%	130%
				(2.30–3.05)		(2.50–3.75)		
** *Other snacks/meals* **								
**Cheese sauce**	*1*	24.18	*2*	7.47	*2*	13.02	224%	86%
				(6.58–8.36)		(10.77–15.27)		
**Croutons**	*1*	42.94	*2*	10.26	*2*	25.84	319%	66%
				(10.00–10.52)		(18.51–33.17)		
**Potato cakes**	*1*	8.68	*2*	3.07	*2*	4.37	183%	99%
				(2.68–3.45)		(1.33–7.41)		
**Potato pots/dehydrated potato**	*3*	87.25	*4*	9.21	*2*	23.46	847%	272%
		(87.25–87.25)		(6.25–12.62)		(20.00–26.93)		
**Soup**	*4*	53.85	*8*	13.03	*4*	26.67	313%	102%
		(48.57–59.18)		(9.26–16.29)		(15.88–34.10)		

Abbreviations: *n* = number of products; SLPFs = special low protein foods. Values displayed as mean (range).

**Table 2 nutrients-13-03977-t002:** Number of units, actual cost of prescribing SLPFs, and percentage of total units and total actual costs of all SLPFs by subgroup from January to December 2020 by the NHS for England.

Subgroup	Number of Units Prescribed from January to December 2020		Actual Costs * from January to December 2020 (£)		For the Year of January to December 2020	
	Total	Monthly Average	Total	Monthly Average	% of Total Units of SLPFs Prescribed	% of Total Actual Cost of SLPFs Prescribed
** *Bread/pizza bases* **						
**Bread (*n* = 12)**	42,171	3514	232,873	19,406	20.2%	10.8%
**Pizza base (*n* = 1)**	3382	282	38,566	3214	1.6%	1.8%
** *Pasta/rice/noodles* **						
**Pasta/rice/noodles (*n* = 33)**	39,043	3254	295,619	24,635	18.7%	13.7%
**Pasta and sauces (prepared) (*n* = 5)**	3574	298	37,592	3133	1.7%	1.7%
**Risotto (*n* = 1)**	258	22	2758	230	0.1%	0.1%
**Xpots (*n* = 4)**	1682	140	36,023	3002	0.8%	1.7%
** *Flour/mixes* **						
**Bread mix (*n* = 1)**	2111	176	11,780	982	1.0%	0.5%
**Cake mix (*n* = 4)**	6790	566	53,697	4475	3.2%	2.5%
**Flour/All Purpose Mix (*n* = 5)**	32,720	2727	239,559	19,963	15.7%	11.1%
**Pancake/waffle mix (*n* = 1)**	700	58	3565	297	0.3%	0.2%
** *Egg replacers* **						
**Egg replacer (*n* = 3)**	1312	109	16,412	1368	0.6%	0.8%
**Egg white replacer (*n* = 1)**	334	28	3398	283	0.2%	0.2%
** *Milk replacers* **						
**Milk replacer (liquid) (*n* = 5)**	*n*/*a*	*n*/*a*	655,437	54,620	n/a	30.5%
**Milk replacer (powder) (*n* = 1)**	*n*/*a*	*n*/*a*	1623	135	n/a	0.1%
** *Meat/fish replacers* **						
**Burger replacements (*n* = 3)**	4601	383	53,038	4420	2.2%	2.5%
**Fish replacement (*n* = 1)**	358	30	4069	339	0.2%	0.2%
**Sausage replacements (*n* = 3)**	7591	633	59,545	4962	3.6%	2.8%
** *Breakfast and cereal bars* **						
**Breakfast bar (*n* = 4)**	1595	133	16,876	1406	0.8%	0.8%
**Breakfast cereal (dried) (*n* = 3)**	6073	506	50,533	4211	2.9%	2.3%
**Fruit bar (*n* = 1)**	6424	535	28,863	2405	3.1%	1.3%
**Hot breakfast cereal (*n* = 4)**	3264	272	27,511	2293	1.6%	1.3%
** *Snacks* **						
**Biscuits/cookies (*n* = 7)**	9841	820	65,126	5427	4.7%	3.0%
**Breadsticks (*n* = 1)**	653	93 **	3928	561 **	0.3%	0.2%
**Cake (*n* = 3)**	3827	319	26,619	2218	1.8%	1.2%
**Chocolate (*n* = 2)**	7299	608	46,714	3893	3.5%	2.2%
**Crackers (*n* = 3)**	12,331	1028	50,952	4246	5.9%	2.4%
**Crisps (*n* = 4)**	1015	85	7528	627	0.5%	0.3%
**Crispbread crackers (*n* = 1)**	180	15	920	77	0.1%	0.0%
**French toast crackers (*n* = 1)**	270	23	1402	117	0.1%	0.1%
**Hazelnut spread (*n* = 1)**	812	68	7219	602	0.4%	0.3%
** *Desserts* **						
**Dessert pot (*n* = 2)**	1548	129	14,782	1232	0.7%	0.7%
**Flavoured pudding (dried) (*n* = 4)**	3188	266	21,439	1787	1.5%	1.0%
**Jelly (dried) (*n* = 2)**	196	16	1728	144	0.1%	0.1%
**Rice pudding (*n* = 4)**	1156	96	7961	663	0.6%	0.4%
**Yogurt substitute (*n* = 1)**	203	17	3855	321	0.1%	0.2%
** *Other snacks/meals* **						
**Cheese sauce (*n* = 1)**	288	24	1716	143	0.1%	0.1%
**Croutons (*n* = 1)**	328	27	2292	191	0.2%	0.1%
**Potato cakes (*n* = 1)**	311	26	2002	167	0.1%	0.1%
**Potato pots (*n* = 3)**	676	56	11,677	973	0.3%	0.5%
**Soup (*n* = 4)**	827	69	4776	398	0.4%	0.2%
**TOTAL**	**208,932**	**17,451**	**2151,973**	**179,566**	**100%**	**100%**

Abbreviations: *n* = number of products; SLPFs = special low protein foods * Actual Costs on ePACT2 is calculated as the Net Ingredient Cost of the item(s) supplied, less the National Average Discount Percentage (NADP) plus Payment for Consumables, Out of Pocket Expenses and Payment for Containers. ** Data from June 2020–December 2020 only.

**Table 3 nutrients-13-03977-t003:** Number of patients in England with PKU and/or using SLPFs under the care of an NHS hospital/centre.

Centre	Number of PKU Paediatric Patients ***	Number of PKU Adult Patients ***	Number of Patients on Full/Partial Phe-Restricted Diet	Number of Non-PKU Inherited Metabolic Disorder Patients Using SLPFs
Birmingham Women’s and Children’s Hospital	110	0	110	15
Evelina London Children’s Healthcare—part of Guy’s and St Thomas’ NHS Foundation Trust	168	0	144	55
Guy’s and St Thomas’ NHS Foundation Trust—Adult IMD service	0	195	145	10
Great Ormond Street Hospital	163	0	159	53
University Hospitals Birmingham NHS Foundation Trust—Queen Elizabeth Hospital	0	153	134	30
University College London Hospitals NHS Foundation Trust	0	378	235	30
Bradford Teaching Hospitals NHS Foundation Trust	58	0	58	21
Royal Manchester Children’s Hospital	96	0	96	27
Bristol Royal Hospital for Children	71	0	67	18
North Bristol NHS Trust	0	58	41	1
Alder Hey Children’s NHS Foundation Trust	54	0	54	17
Salford Royal NHS Foundation Trust	0	334	186	58
Cambridge University Hospitals NHS Foundation Trust	14	36	47—of which 14 are paediatric patients	3
Sheffield Children’s NHS Foundation Trust	52	0	42	21
Sheffield Teaching Hospitals NHS Foundation Trust	0	160	90	20
University Hospitals of Leicester NHS Trust	13	0	10	12
Nottingham University Hospitals NHS Trust	24	0	24	9
Great North Children’s Hospital—within the Royal Victoria Infirmary	64	0	63	9
Royal Victoria Infirmary—Adult IMD services	0	74	43	5
Norfolk and Norwich University Hospital	15	0	15	-
Royal Derby Hospital	6	6	6—all of which are paediatric patients	-
Somerset NHS Foundation Trust	(1)	8 (+1)	5 (+1)	3
Royal Devon & Exeter NHS Foundation Trust	1 (+1)	9	5 (+1)—1 of which is a paediatric patient and not shared care	2
University Hospital Southampton NHS Foundation Trust	4 (+7)	23 (+1)	23 (+8)—4 of which are paediatric patients and not shared care	3
Northamptonshire Healthcare NHS Foundation Trust	10	2	12—10 of which are paediatric patients	0
University Hospitals Bristol & Weston NHS Foundation Trust	0	(21)	(20)	0
**TOTAL**	**923**	**1436**	**1814—877 of which are paediatric patients**	**422**

Abbreviations: SLPFs = special low protein foods; PKU = phenylketonuria; Phe = phenylalanine. ( ) shared care with another unit so numbers not included in totals. *** This includes patients with mild PKU/hyperphenylalaninaemia who maintain phenylalanine levels within target therapeutic range without dietary treatment.

**Table 4 nutrients-13-03977-t004:** NSPKU guideline for recommended amounts of special low protein products per month [19] compared with monthly average per patient estimated in the current study which does not include milk replacers.

Age of Patient with PKU	Recommended Maximum Number of SLPFs to Prescribe Each Month (Not Including Milk Replacers)	Estimated Number of SLPFs Prescribed Per Person Each month (Not Including Milk Replacers)
4 months–3 years	20 units	8 units
4–6 years	25 units	
7–10 years	30 units	
11–18 years	50 units	
Adults	50 units	
Pre-conception/Pregnancy	50 units	

Abbreviations: SLPFs = special low protein foods; PKU = phenylketonuria; NSPKU = The National Society for Phenylketonuria.

## Data Availability

NHSBSA prescribing data on special low protein foods in England was obtained from ePACT2.

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
