# Peer review of "Special Low Protein Foods Prescribed in England for PKU Patients: An Analysis of Prescribing Patterns and Cost"

_nutrients, 2021, doi:10.3390/nu13113977_

Round 1

Reviewer 1 Report

This is a solid study that fills a gap in the literature. The methods are well described with the exception of the following point:

The main edit I would suggest is to include an expanded discussion from the payer perspective. Based on the methods, I think the costs were collected from a formulary or retail costs. Although the authors mention that some patients have out of pocket expenses, the paper doesn't attempt to estimate them, and appears to report all costs as costs to the NHS. Please clarify the methods and/or cost reporting in the results section.

Other specific recommendations for improvement:

  1. Ln 98-99- restructure sentence to define borderline substance at first use
  2. Ln 193- does this analysis assume that all prescribed products were subsequently purchased? If so, is that a reasonable assumption (I'm not familiar with the prescribing system in the UK, but in the US, this would not be the case)? If not, please add to limitations.
  3. Ln 248- provide and example of a SLPF that would require a consumable
  4. Table 4- would move the milk replacement note out of the footnote and into the table title and/or 2nd column header. I missed this on 1st read and it is an important point. Especially for the 4 month to 3 year age group, milk replacements will make a large portion of their diet. This may warrant more discussion.
  5. Ln 305- cite Table 3 here or add to results text. I needed to do some math and digging to figure out where this number came from. 
  6. Ln 326-333- provide context for these costs to allow comparison to your study (ie "approximately £X)
  7. Ln 331- Ref 25 only applies to the beginning of the sentence and would usually be cited earlier in the sentence as well. 

Reviewer 2 Report

This very important study examines the costs of all Special Low Protein Foods (SLPFs) available on prescription in England compared to regular and “free-from “ foods.

The study shows significant differences in the cost of SLFPs compared with the cost of the corresponding regular protein-containing foods and free-from foods, which is understandable from a technological point of view. This results in high annual costs for SLPFs prescribed to patients in the UK (£2,151,973 per year).

The main individuals to whom SLPFs are prescribed are patients with inborn errors of metabolism, with PKU patients being the major consumers. Yet, it has been shown that patients with PKU are prescribed significantly fewer units of SLPFs, if compared to maximum number of units per PKU patients that is recommended by the NSPKU. This applies to both paediatric PKU patients and adults. 

In the interesting discussion, the authors analyzed the various reasons for this situation, especially in relation to adults, who experience different problems with accessing SLPFs. They also raised the problem of different access to SLPFs in different countries.

In conclusion, the authors make some very important recommendations. Firstly, the participation of metabolic dietitians in prescribing SLPFs, as the professionals most involved in the control of dietary treatment of PKU in UK, should be further increased. The authors aptly summarized some limitations on their study.

This is a very good study highlighting the importance of limitations, including organizational ones, in PKU patients' access to SLPFs, which are still the fundamental element of low phenylalanine diet – the gold standard in the treatment of most of PKU patients, even in a highly organized health care system such as the NHS.
